# Metabolic Syndrome, Gamma-Glutamyl Transferase, and Risk of Sudden Cardiac Death

**DOI:** 10.3390/jcm11071781

**Published:** 2022-03-23

**Authors:** Yun Gi Kim, Kyungdo Han, Joo Hee Jeong, Seung-Young Roh, Yun Young Choi, Kyongjin Min, Jaemin Shim, Jong-Il Choi, Young-Hoon Kim

**Affiliations:** 1Division of Cardiology, Department of Internal Medicine, Korea University Anam Hospital, Korea University College of Medicine, Seoul 02841, Korea; tmod0176@gmail.com (Y.G.K.); jessica0115@naver.com (J.H.J.); rsy008@gmail.com (S.-Y.R.); yych60@naver.com (Y.Y.C.); mkj880628@naver.com (K.M.); jaemins@korea.ac.kr (J.S.); yhkmd@unitel.co.kr (Y.-H.K.); 2Department of Statistics and Actuarial Science, Soongsil University, Seoul 06978, Korea; hkd917@naver.com

**Keywords:** gamma-glutamyl transferase, metabolic syndrome, sudden cardiac death

## Abstract

Background: Metabolic syndrome is associated with a significantly increased risk of sudden cardiac death (SCD). However, whether temporal changes in the metabolic syndrome status are associated with SCD is unknown. We aimed to determine whether metabolic syndrome and gamma-glutamyl transferase (ɣ-GTP), including their temporal changes, are associated with the risk of SCD. Methods: We performed a nationwide population-based analysis using the Korean National Health Insurance Service. People who underwent a national health check-up in 2009 and 2011 were enrolled. The influence of metabolic syndrome and ɣ-GTP on SCD risk was evaluated. Results: In 2009, 4,056,423 (848,498 with metabolic syndrome) people underwent health screenings, 2,706,788 of whom underwent follow-up health screenings in 2011. Metabolic syndrome was associated with a 50.7% increased SCD risk (adjusted hazard ratio (aHR) = 1.507; *p* < 0.001). The SCD risk increased linearly as the metabolic syndrome diagnostic criteria increased. The ɣ-GTP significantly impacted the SCD risk; the highest quartile had a 51.9% increased risk versus the lowest quartile (aHR = 1.519; *p* < 0.001). A temporal change in the metabolic syndrome status and ɣ-GTP between 2009 and 2011 was significantly correlated with the SCD risk. Having metabolic syndrome in 2009 or 2011 indicated a lower SCD risk than having metabolic syndrome in 2009 and 2011 but a higher risk than having no metabolic syndrome. People with a ≥20-unit increase in ɣ-GTP between 2009 and 2011 had an 81.0% increased SCD risk versus those with a change ≤5 units (aHR = 1.810; *p* < 0.001). Conclusions: Metabolic syndrome and ɣ-GTP significantly correlated with an increased SCD risk. SCD was also influenced by temporal changes in the metabolic syndrome status and ɣ-GTP, suggesting that appropriate medical treatment and lifestyle modifications may reduce future SCD risk.

## 1. Introduction

Sudden cardiac death (SCD) is an emergent medical condition that requires timely intervention to bring the victim back to life [1,2]. However, since all SCD cases occur outside the hospital, immediate professional intervention is virtually impossible [2,3,4,5]. The SCD survival rates are still not satisfactory, even in the most developed regions of the world, whereas neurologically intact survival rates are even lower [6,7]. The immediate management of SCD is the cornerstone of SCD treatment. However, ensuring the adequate training of the general population for cardiopulmonary resuscitation and the dissemination of automated external defibrillators requires enormous medical resources. Therefore, understanding and meticulously managing the risk factors for SCD require greater attention.

The underlying causes of SCD include coronary artery disease, primary myocardial or electrical disorders such as hypertrophic cardiomyopathy and Brugada syndrome, and valvular heart disease [3,8]. Metabolic syndrome, which is characterized by abdominal obesity, dyslipidemia, an elevated fasting glucose, and high blood pressure, is an established risk factor for coronary artery disease and SCD [9,10]. Hess et al. revealed that the risk of SCD increased by 70% in people with metabolic syndrome [10]. However, the total sample number was not large at 13,168 people with 357 SCD events. Furthermore, the impact of metabolic syndrome management is not fully understood. Gamma-glutamyl transferase (ɣ-GTP) is linked with obesity, physical inactivity, hypertension, dyslipidemia, and glucose intolerance, suggesting it can be a good marker for metabolic syndrome [11,12,13]. Since the presence of metabolic syndrome is associated with an elevated risk of SCD, ɣ-GTP may have predictive value for the occurrence of SCD. However, the impact of ɣ-GTP and changes in ɣ-GTP on SCD remain to be elucidated.

Here, we aimed to evaluate the impact of: (i) metabolic syndrome, (ii) control of the metabolic syndrome, (iii) ɣ-GTP, and changes in ɣ-GTP on the risk of SCD through a nationwide population-based analysis.

## 2. Patients and Methods

### 2.1. Study Design

The Korean National Health Insurance Service (K-NHIS) database was used in this study. Since all citizens of South Korea are mandatory K-NHIS subscribers, its data represents the entire Korean population. The K-NHIS offers a nationwide regular health screening to its subscribers, including measurements of height, weight, waist circumference, and blood pressure; a self-reported questionnaire about alcohol consumption, smoking status, and physical activity level; and various laboratory tests, such as the blood cell count; renal function; liver function; fasting blood glucose; and lipid profile (total cholesterol, high-density lipoprotein, and triglycerides). Prior claims of various International Classification of Disease, tenth edition (ICD-10) diagnostic codes such as diabetes mellitus (DM), hypertension, dyslipidemia, or heart failure, and a prescription history of various drugs, are also recorded in the K-NHIS database. Therefore, the presence of metabolic syndrome can be identified for all K-NHIS subscribers who have undergone the nationwide health screening. The level of ɣ-GTP, a marker for metabolic syndrome, is included in the liver function test [14,15]. It is measured in international units per liter (IU/L), as with prior studies [11,12,13].

This study enrolled patients who underwent a nationwide health screening in 2009. People who were previously diagnosed with SCD and were aged less than 20 years at the 2009 health screening were excluded. Clinical follow-up data were available until December 2018. Since the K-NHIS is a nationwide mandatory and exclusive health care insurance system of the Republic of Korea, there were no follow-up losses, except for immigrants.

To identify baseline medical history variables such as hypertension and DM, the data obtained from January 2002 to December 2008 were used as the screening period, and the robustness of our coding strategy was validated via multiple prior studies [16,17,18,19,20]. Sequential nationwide health screenings are recommended for K-NHIS subscribers. We identified people who underwent a nationwide health screening in 2011 from among those who had undergone a nationwide health screening in 2009. Therefore, sequential alterations in the metabolic syndrome status and ɣ-GTP were available for the analysis.

The current study was approved by the Institutional Review Board of Korea University Medicine Anam Hospital (IRB number: 2021AN0185) and official review committee of the K-NHIS. Considering the retrospective nature of this study, the requirement for written informed consent was waived. The ethical guidelines of the 2013 Declaration of Helsinki and legal medical regulations of the Republic of Korea were strictly undertaken throughout the study.

### 2.2. Primary Outcome Endpoint

The occurrence of SCD was the main outcome of this study. The ICD-10 codes used to identify SCD were as follows: I46.0 (cardiac arrest with successful resuscitation), I46.1 (sudden cardiac arrest), I46.9 (cardiac arrest, cause unspecified), I49.0 (ventricular fibrillation and flutter), R96.0 (instantaneous death), and R96.1 (death occurring less than 24 h from symptom onset). Only claims accompanied by a declaration of death or cardiopulmonary resuscitation were included. In-hospital cardiac arrest was not the scope of this study; therefore, only claims that were coded during an emergency department visit were selected. Not all SCD events were of cardiac origin, and SCD events were not counted as a primary outcome endpoint if the participants had a prior diagnosis of asphyxia, gastrointestinal bleeding, cerebral hemorrhage, ischemic stroke, sepsis, anaphylaxis, trauma, suffocation, lightning strike, electric shock, drowning, or burn within 6 months of the diagnosis of SCD. The incidence of SCD was defined as the number of events per 1000 person/years of follow-up. The impacts of metabolic syndrome, ɣ-GTP, and their alterations over time on SCD incidence were evaluated. The incidence of SCD was defined as the number of events per 1000 person/years of follow-up.

### 2.3. Metabolic Syndrome

The National Cholesterol Education Program Adult Treatment Panel III criteria for metabolic syndrome was used in this study [21]. Metabolic syndrome was defined as the presence of three or more of the following criteria: (i) waist circumference ≥102 cm (men; ≥90 cm for East Asian men) or ≥88 cm (women; ≥80 cm for East Asian women), (ii) blood pressure ≥130/85 mmHg or on pharmacologic treatment, (iii) fasting blood glucose ≥100 mg/dL or on pharmacologic treatment, (iv) fasting triglyceride level ≥150 mg/dL or on pharmacologic treatment, and (v) fasting high-density lipoprotein cholesterol level ≤40 mg/dL (men) or ≤50 mg/dL (women) or on pharmacologic treatment.

### 2.4. Definitions

Diabetes mellitus and impaired fasting glucose were defined based on either the fasting blood glucose (FBG) level (FBG ≥ 126 mg/dL for DM and FBG 100–125 mg/dL for impaired fasting glucose) or a claim of relevant ICD-10 codes by a physician. Hypertension and prehypertension were also identified by measured blood pressure and ICD-10 codes for hypertension and prehypertension. Blood pressure criteria were systolic blood pressure (SBP) <120 mmHg and diastolic blood pressure (DBP) <80 mmHg for non-hypertension, either 120 ≤ SBP < 140 or 80 ≤ DBP < 90 for prehypertension, and either SBP ≥ 140 or DBP ≥ 90 for hypertension. Chronic kidney disease (CKD) was defined as estimated glomerular filtration rate < 60 mL/min/1.73 m^2^, which was calculated by the Modification of Diet in Renal Disease (MDRD) equation. A self-questionnaire acquired during the 2009 health check-up was used to define regular physical activity. People who had one or more sessions in a week with high (such as running, climbing, or intense bicycle activities) or moderate physical activity (such as walking fast, tennis, or moderate bicycle activities) were classified as having regular physical activity. The robustness of the aforementioned definitions was validated in our prior studies [16,17,22].

### 2.5. Statistical Analysis

Continuous variables were expressed as the mean ± standard deviation, and the Student’s *t*-test was used for statistical comparisons. Categorical variables were compared using the chi-square test or Fisher’s exact test, as appropriate. The cumulative incidence of SCD was depicted by a Kaplan–Meier curve analysis, and intergroup differences were compared using the log-rank *t*-test. Raw and adjusted hazard ratios (aHRs) with 95% confidence intervals (CIs) were calculated using a Cox-regression analysis. Since ɣ-GTP was non-normally distributed, it was analyzed as a continuous variable in the Cox-regression analysis. We divided our cohort into quartile groups, and the HR for each group was calculated through a Cox-regression analysis, with the lowest quartile group as a reference group. Bonferroni correction was performed to adjust for the influence of multiple comparisons (if more than three groups are compared to each other). People with missing data were excluded from the study. Statistical significance was defined as *p*-values ≤ 0.05 in two-tailed tests. All statistical analyses were performed using SAS version 9.2 (SAS Institute, Cary, NC, USA).

## 3. Results

### 3.1. Study Population

Among people who underwent nationwide health screening in 2009, 50% of them were randomly selected and a total of 4,234,341 people were enrolled in this analysis. Due to a prior diagnosis of SCD and missing data, 491 and 177,427 people were excluded from the study, respectively. In terms of the baseline demographics (health screening in 2009), 4,056,423 people were analyzed, 2,706,788 among whom underwent a follow-up health screening in 2011. The flow of the study is summarized in Figure 1. The baseline demographics of the patients who did or did not experience SCD are summarized in Table 1. In brief, people who experienced SCD during the follow-up period were significantly older; had larger waist circumferences; were more likely to be male and smokers; had a higher prevalence of hypertension, DM, dyslipidemia, and chronic kidney disease; and had higher ɣ-GTP levels.

### 3.2. Metabolic Syndrome

Baseline characteristics of the people with and without metabolic syndrome are summarized in Table 2. In brief, people with metabolic syndrome were older; had higher body mass index and waist circumference, had a higher prevalence of male sex, hypertension, DM, dyslipidemia, and chronic kidney disease. Among the 848,498 people with metabolic syndrome (with 6,897,608 person/years of follow-up), 6546 SCD events occurred with an incidence of 0.949 cases per 1000 person/years of follow-up. In people without metabolic syndrome, 9,806 SCD events occurred during 26,447,770 person/years of follow-up for an incidence of 0.371 (Table 3). After multivariate adjustment, the metabolic syndrome was associated with a 50.7% increased SCD risk (aHR = 1.507; 95% CI = 1.456–1.560; *p* < 0.001; Table 2). Metabolic syndrome was diagnosed based on the fulfillment of five criteria. The number of criteria fulfilled was linearly associated with the occurrence of SCD, with more criteria being associated with a greater risk (Table 3). People who met all five criteria for metabolic syndrome had a 2.956-fold increased risk of SCD compared with those who met no criteria (aHR = 2.956; 95% CI = 2.707–3.227; *p* < 0.001; Table 3). The SCD risk gradually increased as the number of fulfilled criteria for metabolic syndrome increased (Table 3).

### 3.3. ɣ-GTP

In this nationwide cohort, ɣ-GTP levels were significantly higher in people with metabolic syndrome (median value 23.81, versus 38.86; *p* < 0.001; Table 2). People were classified into four groups according to serum ɣ-GTP concentration. The prevalence of metabolic syndrome in the highest ɣ-GTP level quartile was significantly higher than that in the lowest quartile (9.4% versus 44.1%; *p* < 0.001). Compared with people in the first (lowest ɣ-GTP level) quartile, those in the third (aHR = 1.158; 95% CI = 1.098–1.222; *p* < 0.001; Table 4) and fourth (highest ɣ-GTP level; aHR = 1.519; 95% CI = 1.437–1.605; *p* < 0.001; Table 4) quartiles showed significantly increased risks of SCD.

### 3.4. Temporal Changes

Compared with people without metabolic syndromes in both 2009 and 2011, those who developed metabolic syndrome in 2011 (without metabolic syndrome in 2009) had a 31.6% increased risk of SCD (aHR = 1.316; 95% CI = 1.215–1.426; *p* < 0.001; Table 5; Figure 2). People who had metabolic syndrome in both 2009 and 2011 had a 59.1% increased risk of SCD (aHR = 1.591; 95% CI = 1.503–1.684; *p* < 0.001; Table 5 and Figure 2). However, the SCD risk was significantly lower if metabolic syndrome recovered in 2011 (aHR = 1.310; 95% CI = 1.207–1.423; *p* < 0.001; Table 5 and Figure 2).

Temporal changes in the ɣ-GTP level were also associated with SCD risk. Compared with people who maintained a change of the ɣ-GTP level within five units, people who had a more than 20-unit increase between 2009 and 2011 had an 81.0% increased SCD risk (aHR = 1.810; 95% CI = 1.683–1.947; *p* < 0.001; Figure 2). People with a change in the ɣ-GTP level of 10–20 units showed a 19.0% increased SCD risk (aHR = 1.190; 95% CI = 1.094–1.294; *p* < 0.001; Figure 2).

## 4. Discussion

This nationwide cohort-based study demonstrated that (i) metabolic syndrome was associated with a significantly increased SCD risk, (ii) the more number of criteria for metabolic syndrome fulfilled, the higher the SCD risk, (iii) the successful management of metabolic syndrome was associated with an alleviated SCD risk, (iv) people with uncontrolled metabolic syndrome were at higher SCD risk, (v) an increased ɣ-GTP level was significantly associated with an increased SCD risk, and (vi) a temporal increase in the ɣ-GTP level was significantly associated with an increased SCD risk.

Metabolic syndrome is an increasingly prevalent medical condition. This study revealed that metabolic syndrome, especially when uncontrolled, increases one’s SCD risk. Furthermore, the successful management of metabolic syndrome may reduce the risk of SCD, indicating the importance of lifestyle modifications and medical treatment in people with metabolic syndrome. The prognosis of people with SCD is poor, and its primary prevention is especially important considering the intrinsic limitations in its management [6,7,23,24]. Our results can provide important clues for reducing SCD-related healthcare burdens.

### 4.1. Metabolic Syndrome

Metabolic syndrome is a cluster of central obesity, dyslipidemia, glucose intolerance, and elevated blood pressure [25]. Empana et al. reported a 68% increased risk of sudden death attributable to metabolic syndrome in middle-aged men [26]. Hess et al. also reported a 70% increased risk of sudden cardiac death among participants of the ARIC (Atherosclerosis Risk in Communities) study [10]. Our results are in accordance with those of the aforementioned studies, and our participants with metabolic syndrome showed a 50.7% increased risk of SCD. However, the current study is distinguished by the following points: (i) it included a significantly large number of participants, (ii) we simultaneously analyzed ɣ-GTP, and (iii) we analyzed the temporal changes in metabolic syndrome. The data of 4,056,423 people were analyzed, 848,498 of whom had metabolic syndrome. The total number of SCD events was 16,352, which was sufficient for various analyses. We divided our participants into six groups according to the number of metabolic syndrome criteria fulfilled and revealed that the SCD risk was the highest among those who met all five criteria. A linear increase in the SCD risk was noted as the number of fulfilled criteria increased, suggesting that there are no clear cut-off criteria, but rather, every aspect of metabolic syndrome matters. The clinical impact of temporal changes in the metabolic syndrome status is another strong aspect of this study. People with metabolic syndrome in the 2009 and 2011 nationwide health screenings were at the highest risk of SCD. People with newly developed metabolic syndrome in 2011 showed a 31.6% increased risk of SCD compared to those without metabolic syndrome in 2009 or 2011. Importantly, people who did not have metabolic syndrome in 2011 were at a significantly lower risk of developing SCD than those who had metabolic syndrome in both 2009 and 2011, suggesting that the adequate management of metabolic syndrome can have a significant therapeutic effect on reducing SCD risk.

### 4.2. Gamma-GTP

The serum ɣ-GTP level is routinely measured in nationwide health screening programs in South Korea; here, we were able to analyze large amounts of ɣ-GTP data. People with the highest ɣ-GTP level quartile were at a 51.9% increased risk of SCD compared with those in the lowest quartile. The association between ɣ-GTP and metabolic syndrome was described in prior studies of participants with higher ɣ-GTP levels showing a significantly higher risk of metabolic syndrome [14,27]. However, the association between ɣ-GTP level and SCD has not been described until now. The highest quartile group in our study, by definition, comprised 25% of the general population, and these people demonstrated a 51.9% increased SCD risk. This group vulnerable to SCD is not a small specific subgroup, suggesting that the ɣ-GTP level can be a very useful serum marker for identifying people at high risk for SCD. Furthermore, we demonstrated here that a temporal increase in the ɣ-GTP level of more than 20 units was associated with an 81.0% increased risk of developing SCD.

Insulin resistance is the suspected underlying mechanism linking ɣ-GTP and metabolic syndrome [14,27]. The prevalence of metabolic syndrome, number of its components, and insulin resistance increased as the ɣ-GTP level increased in a prior study of 3246 Korean people [28]. Kawamoto et al. reported that the association between ɣ-GTP and metabolic syndrome was significantly attenuated after the adjustment for markers of insulin resistance, suggesting that such an association is correlated with insulin resistance [14]. Insulin resistance is also the underlying pathophysiology of metabolic syndrome [29], and it may be the cause of the increased risk of SCD developing among the people with metabolic syndrome and high ɣ-GTP levels in this study. Whether improvements in insulin resistance can decrease the SCD risk will be an important area of future research.

An increased level of ɣ-GTP might also reflect the degree of inflammation, an important pathophysiology of atherosclerosis, explaining the potential association between high ɣ-GTP and increased risk of SCD observed in our study [14,30]. A prior study revealed that ɣ-GTP can increase low-density lipoprotein oxidation through hydrolyzing extracellular glutathione into more potent iron reductants, another possible mechanism linking ɣ-GTP, atherosclerosis, and SCD [31].

### 4.3. Limitations

This study has several limitations. Despite prior validation of our coding strategies, coding inaccuracies can arise from retrospective analyses of nationwide health insurance organization data [16,20,32,33]. The use of an exclusive cohort, which consisted solely of East Asian people, limited the generalizability of our findings. Additionally, liver function can affect the serum level of ɣ-GTP. However, we were not able to obtain liver function tests, such as the liver enzyme level, bilirubin level, alkaline phosphatase, or prothrombin time. Finally, we were able to assess the occurrence but not the result of SCD development.

## 5. Conclusions

The risk of SCD is significantly increased in people with metabolic syndrome, and the number of fulfilled metabolic syndrome diagnostic criteria and temporal changes in its status are significantly associated with the risk of SCD. The baseline level and temporal changes in ɣ-GTP, a serum marker for metabolic syndrome, are also significantly associated with the future risk of SCD. Our findings indicate that efforts to control metabolic syndrome and ɣ-GTP levels can reduce the risk of SCD.

## Figures and Tables

**Figure 1 jcm-11-01781-f001:**
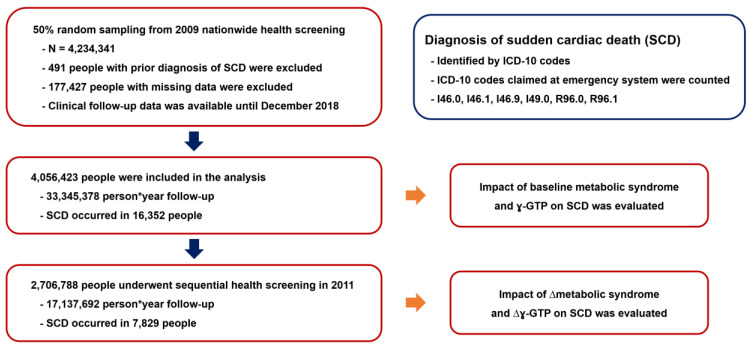
Study flow. ɣ-GTP: gamma-glutamyl transferase; ICD-10: International Classification of Disease, tenth edition; SCD: sudden cardiac death.

**Figure 2 jcm-11-01781-f002:**
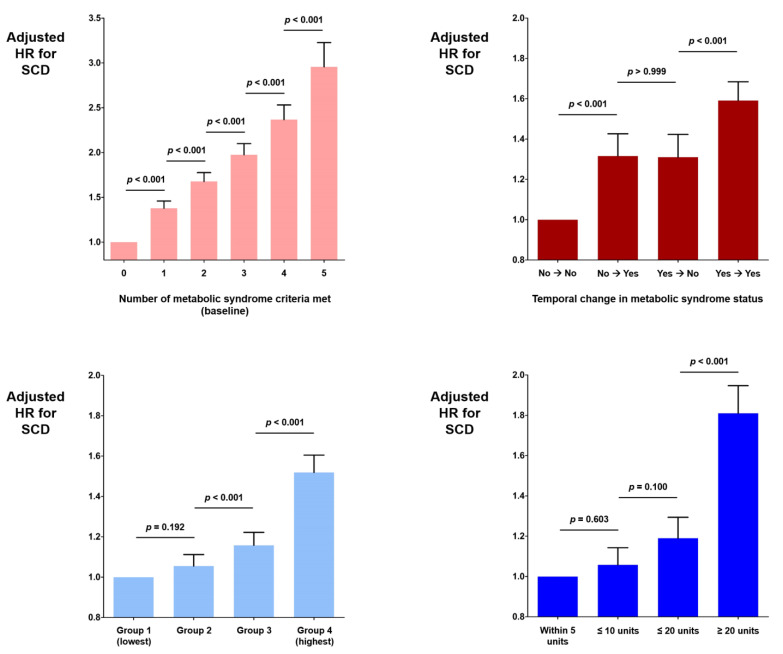
Impact of metabolic syndrome and ɣ-GTP on SCD. The risk of SCD increased linearly according to the number of metabolic syndrome criteria met and serum ɣ-GTP level (**left panels**). Temporal changes in the metabolic syndrome status and serum level of ɣ-GTP significantly influenced the SCD risk (**right panels**). The whiskers indicate 95% confidence intervals. *p*-values are presented with Bonferroni correction. ɣ-GTP, gamma-glutamyl transferase; HR, hazard ratio; SCD, sudden cardiac death.

**Table 1 jcm-11-01781-t001:** Baseline characteristics of SCD victims.

	SCD		*p*-Value
	No	Yes	
	4,040,071	16,352	
Male sex	2,221,898 (55.0%)	11,633 (71.1%)	<0.001
Age (year)	47.0 ± 14.1	62.0 ± 13.2	<0.001
Body mass index (kg/m^2^)	23.7 ± 3.2	23.8 ± 3.4	0.138
Waist circumference (cm)	80.2 ± 9.5	83.5 ± 8.9	<0.001
Smoking history			<0.001
Never-smoker	2,399,679 (59.4%)	7916 (48.4%)	
Ex-smoker	581,485 (14.4%)	3128 (19.1%)	
Current-smoker	1,058,907 (26.2%)	5308 (32.5%)	
Alcohol consumption			<0.001
Non-drinker	2,077,053 (51.4%)	9534 (58.3%)	
Mild-drinker	1,641,427 (40.6%)	5263 (32.2%)	
Heavy-drinker	321,591 (8.0%)	1555 (9.5%)	
Regular Exercise	733,609 (18.2%)	3148 (19.3%)	<0.001
Income (lowest 20% group)	704,587 (17.4%)	3075 (18.8%)	<0.001
Diabetes mellitus	349,134 (8.6%)	4264 (26.1%)	<0.001
Serum glucose (mg/dL)	97.2 ± 23.8	110.0 ± 41.5	<0.001
Hypertension	1,082,382 (27.0%)	9331 (57.1%)	<0.001
Systolic blood pressure (mmHg)	122.4 ± 15.0	129.3 ± 17.2	<0.001
Diastolic blood pressure (mmHg)	76.3 ± 10.0	78.9 ± 11.0	<0.001
Dyslipidemia	732,983 (18.1%)	4610 (28.2%)	<0.001
Cholesterol (mg/dL)	195.3 ± 41.1	195.1 ± 44.3	0.549
High density lipoprotein (mg/dL)	56.5 ± 32.9	53.6 ± 30.9	<0.001
Low density lipoprotein (mg/dL)	121.2 ± 214.2	115.0 ± 97.8	<0.001
Chronic kidney disease	275,854 (6.8%)	2740 (16.8%)	<0.001
eGFR (mL/min/1.73 m^2^)	87.6 ± 44.9	80.4 ± 34.7	<0.001

eGFR: estimated glomerular filtration rate; SCD: sudden cardiac death.

**Table 2 jcm-11-01781-t002:** Baseline characteristics of patients with versus without metabolic syndrome.

	Metabolic Syndrome		*p* Value
	No	Yes	
	3,207,925	848,498	
Male sex	1,717,481 (53.6%)	516,050 (60.8%)	<0.001
Age	45.0 ± 13.6	55.0 ± 12.9	<0.001
Age group, years			<0.001
20–29	480,147 (15.0%)	21,469 (2.5%)	
30–39	687,795 (21.4%)	91,015 (10.7%)	
40–49	898,205 (28.0%)	167,654 (19.8%)	
50–59	627,768 (19.6%)	233,180 (27.5%)	
60–69	337,250 (10.5%)	211,238 (24.9%)	
70–79	152,342 (4.8%)	109,341 (12.9%)	
80+	24,418 (0.8%)	14,601 (1.7%)	
Body mass index	23.1 ± 2.9	26.0 ± 3.2	<0.001
Waist circumference	78.3 ± 8.5	87.9 ± 9.0	<0.001
Smoking status			<0.001
Non-smoker	1,940,944 (60.5%)	466,651 (55.0%)	
Ex-smoker	428,522 (13.4%)	156,091 (18.4%)	
Current smoker	838,459 (26.1%)	225,756 (26.6%)	
Alcohol consumption			<0.001
Non-drinker	1,639,224 (51.1%)	447,363 (52.7%)	
Mild drinker	1,341,119 (41.8%)	305,571 (36.0%)	
Heavy drinker	227,582 (7.1%)	95,564 (11.3%)	
Regular exercise	569,089 (17.7%)	167,668 (19.8%)	<0.001
Income (lower 20%)	564,051 (17.6%)	143,611 (16.9%)	<0.001
Diabetes mellitus	117,925 (3.7%)	235,473 (27.8%)	<0.001
Diabetes mellitus stage			<0.001
Non-diabetic	2,541,796 (79.2%)	242,335 (28.6%)	
Impaired fasting glucose	548,204 (17.1%)	370,690 (43.7%)	
New-onset	53,175 (1.7%)	67,408 (7.9%)	
<5 years	32,167 (1.0%)	87,320 (10.3%)	
≥5 years	32,583 (1.0%)	80,745 (9.5%)	
Fasting blood glucose (mg/dL)	92.9 ± 17.7	113.5 ± 34.6	<0.001
Hypertension	536,787 (16.7%)	554,926 (65.4%)	<0.001
Hypertension stage			<0.001
Non-hypertensive	1,340,278 (41.8%)	45,699 (5.4%)	
Pre-hypertension	1,330,860 (41.5%)	247,873 (29.2%)	
Hypertension without medication	210,930 (6.6%)	125,149 (14.8%)	
Hypertension with medication	325,857 (10.2%)	429,777 (50.7%)	
Systolic blood pressure	119.8 ± 14.0	132.4 ± 14.7	<0.001
Diastolic blood pressure	74.9 ± 9.5	81.7 ± 10.1	<0.001
Dyslipidemia	300,945 (9.4%)	436,648 (51.5%)	<0.001
Dyslipidemia stage (mg/dL)			<0.001
Total cholesterol < 240	2,906,980 (90.6%)	411,850 (48.5%)	
Total cholesterol ≥ 240	257,907 (8.0%)	90,765 (10.7%)	
Total cholesterol ≥ 240 with medication	43,038 (1.3%)	345,883 (40.8%)	
Cholesterol level (mg/dL)	192.3 ± 38.4	206.5 ± 48.5	<0.001
High-density lipoprotein (mg/dL)	57.5 ± 31.8	52.8 ± 36.7	<0.001
Low-density lipoprotein (mg/dL)	122.1 ± 232.1	117.8 ± 122.1	<0.001
Chronic kidney disease	182,472 (5.7%)	96,122 (11.3%)	<0.001
Estimated glomerular filtration rate	88.8 ± 46.6	83.0 ± 37.4	<0.001
Gamma-glutamyl transferase (IU/L)	23.81 (23.79-23.82)	38.86 (38.80-38.93)	<0.001

Data are shown as n (%), mean ± standard deviation, or median (95% confidence interval). IU/L: international units per liter.

**Table 3 jcm-11-01781-t003:** Impact of metabolic syndrome on SCD.

	n	SCD	Follow-up Duration (Person/Years)	Incidence	Hazard Ratio with 95% Confidence Interval	
					Univariate	Multivariate
Metabolic syndrome						
No	3,207,925	9806	26,447,770	0.371	1 (reference)	1 (reference)
Yes	848,498	6546	6,897,608	0.949	2.558 (2.480–2.640)	1.507 (1.456–1.560)
Number of metabolic syndrome criteria met						
0	1,261,043	1806	10,465,831	0.173	1 (reference)	1 (reference)
1	1,127,888	3819	9,285,122	0.411	2.383 (2.254–2.521)	1.378 (1.302–1.459)
2	818,994	4181	6,696,816	0.624	3.619 (3.424–3.824)	1.677 (1.583–1.776)
3	509,764	3378	4,153,728	0.813	4.712 (4.451–4.990)	1.975 (1.858–2.100)
4	259,869	2287	2,107,073	1.085	6.288 (5.911–6.688)	2.367 (2.214–2.531)
5	78,865	881	636,807	1.383	8.008 (7.388–8.680)	2.956 (2.707–3.227)

Incidence is per 1000 person/years of follow-up. SCD: sudden cardiac death. The multivariate model was adjusted for age, sex, body mass index, smoking status, alcohol consumption, regular physical activity, and income level.

**Table 4 jcm-11-01781-t004:** Impact of ɣ-GTP level on SCD development.

	n	SCD	Follow-Up Duration (Person/Years)	Incidence	Hazard Ratio with 95% Confidence Interval	
					Univariate	Multivariate
ɣ-GTP						
Q1 (lowest)	987,003	2314	8,162,351	0.284	1 (reference)	1 (reference)
Q2	1,066,117	3727	8,789,346	0.424	1.496 (1.420–1.575)	1.055 (1.000–1.112)
Q3	996,270	4447	8,184,478	0.543	1.918 (1.824–2.017)	1.158 (1.098–1.222)
Q4 (highest)	1,007,033	5864	8,209,202	0.714	2.528 (2.409–2.652)	1.519 (1.437–1.605)

Incidence is per 1000 person/years of follow-up. ɣ-GTP, gamma-glutamyl transferase; SCD, sudden cardiac death, the multivariate model was adjusted for age, sex, body mass index, waist circumference, smoking status, alcohol consumption, regular physical activity, income level, hypertension, diabetes mellitus, dyslipidemia, and chronic kidney disease.

**Table 5 jcm-11-01781-t005:** Impact of changes in metabolic syndrome status on SCD development.

	n	SCD	Follow-Up Duration (Person/Years)	Incidence	Hazard Ratio with 95% Confidence Interval	
					Univariate	Multivariate
Metabolic syndrome						
No (in 2009) → No (in 2011)	1,967,483	4,080	12,479,939	0.327	1 (reference)	1 (reference)
No → Yes	189,148	733	1,193,749	0.614	1.877 (1.735–2.030)	1.316 (1.215–1.426)
Yes → No	169,640	702	1,068,427	0.657	2.009 (1.854–2.176)	1.310 (1.207–1.423)
Yes → Yes	380,517	2,314	2,395,577	0.966	2.946 (2.800–3.101)	1.591 (1.503–1.684)

Incidence is per 1000 person/years of follow-up. SCD: sudden cardiac death. The multivariate model was adjusted for age, sex, body mass index, smoking status, alcohol consumption, regular physical activity, and income level.

## Data Availability

The raw data underlying this article cannot be shared publicly due to privacy reasons and legal regulations of the Republic of Korea. The raw data is stored and analyzed only in the designated server managed by the K-NHIS.

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
