# Peer review of "Metabolic Syndrome, Gamma-Glutamyl Transferase, and Risk of Sudden Cardiac Death"

_jcm, 2022, doi:10.3390/jcm11071781_

Round 1

Reviewer 1 Report

Authors presented a manuscript showing the association between metabolic syndrome (MS), GGT and sudden cardiac death (SCD). The topic is interesting and provide data on population from Korea.

However there are several issues that should be clarified:

  • The choice of diagnostic criteria should be provided (WHO/ATP III/IDF/AHA?). Authors should also decide whether the cut-off points for the studied population were selected correctly
  • In the methods there are cm in waist circ, but those should be inches – I suppose
  • I am lacking the inclusion of current treatment in the diagnosis of MS (was it included?)
  • I am also lacking the definition of chronic kidney disease in this particular study
  • Table S1 should be included in the manuscript not as supplement. There are a lot of confounders that reader should be aware of
  • While the MS and SCD data seem interesting, the GGT may be flawed by the lack of ALT, AST, bilirubin and alkaline phosphatase, which are essential in this case. I suggest omitting the data regarding GGT
  • GGT was shown not to be distributed normally, but the statistical analysis section lacks the information regarding the method of comparisons in this case
  • GGT level was provided in “units”?? What kind of “units”??
  • In the temporal changes of patient who were freed from MS during follow up authors state that the risk of SCD is reduced: “People who had metabolic syndrome in both 2009 and 2011 had a 59.1% increased risk of SCD (aHR = 1.591; 95% CI = 1.503–1.684; p < 0.001; Table 4; Figure 2). However, the SCD risk was significantly lower if metabolic syndrome recovered in 2011 (aHR = 1.310; 95% CI = 1.207–1.423; p < 0.001; Table 4; Figure 2).” However no direct comparison has been shown that include p value
  • Table 2 is located in front of table 1 in the manuscript
  • 2 lacks clear depiction of statistically significant differences between the groups. These are to some extent in tables that would greatly add to the clarity
  • Finally authors should firmly state whether the source data are available or not. Currently it is ambiguous

Author Response

Dear reviewers,

We appreciate your time and effort to provide insightful comments and suggestions that are kindly aimed to improve the quality of our manuscript. Your comments allowed us to view our own data from a different perspective and elaborate our analysis, which was an intellectually challenging experience. Point-by-point responses have been presented below each comment and included in the revised manuscript. Before you proceed to our responses to your comments, we would like to inform you that new contents included in the revised manuscript are highlighted by yellow background color for your convenience. We hope our responses would meet your requirements and expectation.

Sincerely yours,

Jong-Il Choi, MD, PhD, MHS.

Reviewer 2 Report

Here authors describe the association of Gamma-glutamyl Transferase and Risk of Sudden Cardiac Death. However manuscript can be improve

If existing literature about clinical studies related to Gamma-glutamyl Transferase and other cardiovascular disease. In addition, its important to discuss suggested molecular mechanism of increase in Gamma-glutamyl Transferase in metabolic and cardiovascular diseases.

Author Response

Here authors describe the association of Gamma-glutamyl Transferase and Risk of Sudden Cardiac Death. However, manuscript can be improved

#1. If existing literature about clinical studies related to Gamma-glutamyl Transferase and other cardiovascular disease. In addition, its important to discuss suggested molecular mechanism of increase in Gamma-glutamyl Transferase in metabolic and cardiovascular diseases.

  • We fully appreciate your time and effort to review our manuscript.
  • As with your recommendation, we intensified our discussion on possible underlying mechanism linking É£-GTP and SCD. Potential candidates are insulin resistance, inflammation, and LDL oxidation. Modified discussion are as follows.
  • Insulin resistance is the suspected underlying mechanism linking É£-GTP and metabolic syndrome [14,29]. The prevalence of metabolic syndrome, number of its components, and insulin resistance increased as É£-GTP level increased in a prior study of 3,246 Korean people [30]. Kawamoto et al. reported that the association between É£-GTP and metabolic syndrome was significantly attenuated after the adjustment for markers of insulin resistance, suggesting that such an association is correlated with insulin resistance [14]. Insulin resistance is also the underlying pathophysiology of metabolic syndrome [31], and it may be the cause of the increased risk of SCD developing among the people with metabolic syndrome and high É£-GTP levels in this study. Whether improvements in insulin resistance can decrease the SCD risk will be an important area of future research. Increased level of É£-GTP might also reflect the degree of inflammation, an important pathophysiology of atherosclerosis explaining the potential association between high É£-GTP and increased risk of SCD observed in our study [14,32]. Prior study revealed that É£-GTP can increase low-density lipoprotein oxidation through hydrolyzing extracellular glutathione into more potent iron reductants, another possible mechanism linking É£-GTP, atherosclerosis, and SCD [33]. [page 15, line 17 – 22]

Sincerely yours,

Round 2

Reviewer 1 Report

Authors presented a reviewed version of manuscript.

The manuscript is somewhat improved, but the merit is still missing. 

  1. My major concern is the lack of other than GGt markers of liver functiom.
  2. The statstical analysis description was improved, the p value was added to Fig. 2. Unfortunately, according to the description there were no corrections for multiple comparisons
  3. The MS definition is not the best solution in Asian population
  4. CKD definition was provided, but eGFR estimation formula/model was not provided
